materials science/computational chemistry

molecular dynamics simulations, plasticization, solid propellant, combustion

**Author for correspondence:**
Lilong Yang
e-mail: yanglilong.1988@163.com

# Structure and property of propellant based on nitroglycerine/glycerol triacetate mixed plasticizers: molecular dynamics simulation and experimental study

Lilong Yang, Xionggang Wu, Junqiang Li, Tao Chen, Meng Liu and Qiwen He

Xi'an Modern Chemistry Research Institute, Xi'an 710065, People's Republic of China

LY, 0000-0002-1418-2744

Molecular dynamics simulation has been used to investigate the influence of nitroglycerine (NG)/glycerol triacetate (GTA) mixed plasticizers on the plasticizing ability of nitrocellulose (NC) binder in solid propellant. The radial distribution function and binding energy indicated that NC/plasticizers blends showed stronger intermolecular interaction of van der Waals and hydrogen bonds. The mean-squared displacements of plasticizers and volume distribution revealed that the mobility of plasticizer GTA in the NC polymer binder was higher than that of NG. Then, the mechanical properties of the propellant based on NG/GTA mixed plasticizers were investigated systematically using experimental and simulation calculation method. The results suggested that the ductility of propellant based on NG/GTA mixed plasticizers was improved, implying that NG/GTA mixed plasticizers have a higher plasticizing efficiency for NC. Furthermore, we conducted experimental studies on the effects of NG/GTA mixed plasticizers on the energy and combustion properties of propellants. It was shown that NG/GTA mixed plasticizers could enhance the combustion efficiency of propellants effectively at low pressures. These computational and experimental studies provided guidance for the application of NG/GTA mixed plasticizers in high-performance propellants.

# 1. Introduction

As the essential energy source of weapon systems, solid propellants are widely applied to medium-caliber rockets and various gas generators [1–3]. Conventional composite modified double-base (CMDB) propellants comprise binder nitrocellulose (NC), plasticizer nitroglycerine (NG), energetic ingredients nitramine (like hexogeon RDX and octogen HMX) and other minor additives such as processing aids [4,5]. The simple composition, high energy, environment-friendliness and low smoke signature are some of the main advantages of CMDB propellant [6,7]. The binder as the main body and skeleton of a propellant is an important high-performance component, which glues together other additives to obtain excellent overall properties, such as dimensional stability, structural integrity and mechanical property [8–10]. NC obtained by nitration of cellulose is widely applied to CMDB propellants as binder [11]. NC used as binder possesses properties such as good chemical compatibility with CMDB propellant and other components, excellent dispersion uniformity, wide range of raw materials and lower production costs [12–14]. Plasticizer is a type of material for improving the plasticity and processing ability of the propellant, as well as flexibility and durability. Generally, the plasticizing ability of plasticizer affects the performances of combustion, mechanics and energy of propellants [15,16].

Many different types of plasticizers have been used in propellant formulations, but they need to meet some specifications regarding the final application. There has been a significant amount of low viscosity nitrogenous compounds being developed in recent years including N-butyl-N-(2-nitroxyethyl) nitramine (Bu-NENA), trimethylolethane trinitrate (TMETN), butanetriol trinitrate (BTTN) and bis-(2,2-dinitropropyl)-acetal/formal (A3, BDNPA/F). These compounds as plasticizers possess superior ability to enhance the performance of propellant such as mechanics, energy and safety [17–21]. Yang *et al*. [19] reported their study on the properties for the different NC and glycidyl azide polymer (GAP) matrix composites with an azido nitrate ester plasticizer (2-azidoethyl nitrate ester, AENE) by theoretical simulation, in which the addition of AENE was observed to enhance the mechanical property of NC and GAP. In addition to improving the energy performance of propellant, the volatility of energetic groups such as $NO_2$ (nitro group), $N$-$NO_2$ (nitroamine group) and $N_3$ (azide group) in nitrogen-containing plasticizers is unacceptable. This is the reason that the propellant is more vulnerable when exposed to mechanical load and temperature [22–24]. NG as the most commonly used plasticizer for NC exhibits a much lower vaporization rate and higher melt point, which easily results in poor mechanical properties of propellants and causes unstable combustion, even potentially explosion accidents [25]. To avoid problems associated with NG, our research focused on improving the safety performance at low temperatures of CMDB propellants. In this regard, glycerol triacetate (GTA) is particularly interesting as it is nontoxic, non-volatile and has a lower tendency to migrate to the polymer surface during propellant processing and storage, showing an excellent plasticizing ability to NC [26,27]. The GTA plasticizer increases the chain flexibility and decreases the glass transition temperature ($T_g$) for NC [28,29]. The use of mixed plasticizers is the most efficient way, which can improve the mechanical and safety properties of propellant at low temperatures.

The development of propellant technology has allowed the capability to obtain higher energy and less vulnerable composition [30]. A low-vulnerability propellant was prepared by using cellulose acetate (CA) and hydroxy terminated polybutadiene (HTPB) as binders and GAP with molecular weight of 400–600 as a high-energy plasticizer, and its insensitivity, combustion and mechanical performance was investigated [31]. The ballistic, mechanical and thermal performance of RDX-based LOVA propellant was reported in detail by Kumari *et al*. [32]. Above-mentioned propellant was prepared by using CA and NC combination as binder and novel tetraazido ester instead of non-energetic triacetin as plasticizer. The interaction between plasticizers and binder was neglected while numerous studies have focused on improving the insensitivity, mechanical and thermal properties of propellants [33,34]. In this paper, molecular dynamics (MD) simulations were performed to investigate the influence of NG/GTA mixed plasticizers on the plasticizing ability of NC binder in solid propellant. The intermolecular interaction between NC and plasticizers has been evaluated using the radial distribution function and the binding energy. In addition, the mean-squared displacements (MSDs) of plasticizers and volume distribution were used to explore the migration ability of the plasticizer in NC. Besides, plasticizing efficiency of NG/GTA mixed plasticizers on NC was analysed by mechanical properties determined from the MD simulation and experimental study. Furthermore, we conducted experimental studies on the effects of NG/GTA mixed plasticizers on the energy and burning performance of CMDB propellant. The ideal energetic performance of propellant was obtained by theoretical calculation based on the principle of minimum free energy. The burning

**Figure 1.** Molecular (*a*) and optimized chemical structures (*b*) of the NC polymer and NG and GTA plasticizers.

rates and combustion wave structure of propellants were investigated. It is believed that the computational and experimental study is able to increase the understanding of the interaction and mechanisms between a NC polymer and NG/GTA mixed plasticizers in solid propellants, and provide guidance for the applications of NG/GTA mixed plasticizers in high-performance propellants.

# 2. Material and method

## 2.1. Construction of models

MD simulations were carried out using Material Studio 8.0 software. The influence of NG/GTA mixed plasticizers on the plasticizing ability of NC binder was studied by COMPASS (Condensed-phase Optimized Molecular Potential for Atomistic Simulation Study) force field. Previously, the COMPASS force field was used in molecular simulation of polymer [35]. The Visualizer module of Materials Studio software was applied to create molecular models of NC, NG and GTA, and then the structure was optimized by the Forcite module. The Homopolymer Builder module was used to construct the NC polymer chain with 20 repeat units and nitrogen contents of 12%. The molecular and optimized chemical structures of NC, NG and GTA are presented in figure 1. And here, carbon atoms, hydrogen atoms, nitrogen atoms and oxygen atoms are represented as grey, white, blue and red, respectively. In order to facilitate subsequent analysis, the atoms are labelled according to the different type of atomic force field. O atoms of the hydroxyl group for NC are assigned to $O_1$, O and N atoms of the nitro group for NG are labelled $O_2$ and $N_1$, respectively, O atoms attached to nitro group in NG as $O_3$, O atoms in the carbonyl group in GTA are denoted to $O_4$, and O atoms attached to carbonyl group of GTA as $O_5$ in figure 1*b*.

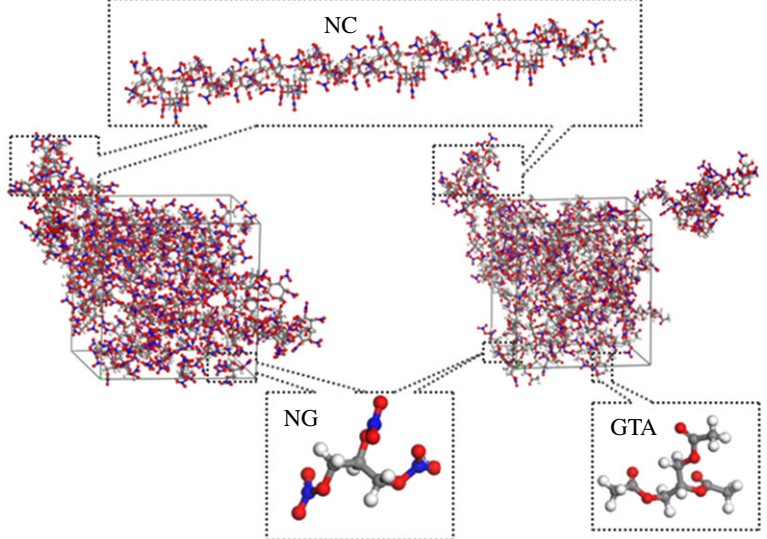

**Figure 2.** Molecular models of NC/NG and NC/NG/GTA blends.

## 2.2. Molecular dynamics simulation method

For investigation of the influence of NG/GTA mixed plasticizers on the plasticizing ability of NC binder, pure NC and NC/NG and NC/NG/GTA blends were constructed by Amorphous cell module, as presented in figure 2. The weight proportion of NC and plasticizer was nearly 50 : 50, and the mass ratios of NG and GTA in mixed plasticizers were nearly 3 : 1, 2 : 1, 1 : 1, respectively. Simulation procedure was conducted by Forcite module under COMPASS force field. To begin with, geometry optimization of the amorphous systems such as pure NC and NC/NG and NC/NG/GTA blends was conducted by Smart Algorithm method minimization with 10 000 steps and convergence level of fine in order to get the eventual structure with minimum energy. Afterwards, 400 ps MD simulations were executed on individual systems to get the equilibrium conformations at the isothermal-isobaric (NPT) ensemble. The parameters were set such that the Andersen thermostat temperature was 298 K and Berendsen barostat pressure was 101.325 kPa, and the time step was assigned as 1 fs. An atom-based method and Ewald summation were used for van der Waals and electrostatic interactions, respectively. The MD simulation system reached equilibrium when the energy fluctuated slightly near the average energy value or the temperature fluctuation was within 10 K. Consequently, the influence of NG/GTA mixed plasticizers on the structure, energy and mechanical performance of NC was analysed to obtain the radial distribution function, intermolecular interaction energy, MSD, volume distribution and static elastic properties.

## 2.3. Materials and propellant preparation

The CMDB propellant was composed of NC, GTA, NG, RDX, combustion catalyst and other additive reagents. The NC/NG/GTA blends are used as energetic binder and plasticizer system to ensure that the propellants achieve higher performance, excellent structural integrity and lower vulnerability. GTA was bought from Shanghai Lingfeng Chemical Reagent Co. Ltd (China). NC was obtained from Sichuan Nitrocell Co. Ltd (China). NG was provided by Xi'an Modern Chemistry Research Institute (China). Hexogeon (RDX) was obtained from Gansu Yinguang Chemical Industrial Group Co. Ltd (China). Four different propellant formulations with different content of NG and GTA were designed (table 1) and prepared. The preparation of propellant includes three basic steps, namely granule preparation, casting and solidification. Above all, the mixture containing the whole NC, solid components and a part of the plasticizer was fabricated into right circular cylinder granules with diameter and length of about 1 mm. Afterwards, the granules were put into a mould, and the interstitial spaces among the granules were filled with a casting solvent composed of residual plasticizer. Finally, under heating, the plasticizer diffused into the polymer binder to solidify the two parts into an integral solid propellant.

**Table 1.** Main ingredients of the CMDB propellants.

| components | content (wt %) | | | |
|---|---|---|---|---|
| | F-1 | F-2 | F-3 | F-4 |
| NC | 22.35 | 22.35 | 22.35 | 22.35 |
| RDX | 38 | 38 | 38 | 38 |
| NG | 30 | 22.5 | 20 | 15 |
| GTA | — | 7.5 | 10 | 15 |
| catalyst and other additives | 9.65 | 9.65 | 9.65 | 9.65 |

## 2.4. Properties evaluation of propellant

Tensile testing was carried out on an Instron 4505 (USA) universal testing machine with a temperature of −40°C, 20°C and 50°C. The cured propellant sample was cut into slices with thickness of 2 mm, and the slice was stamped into a dumbbell shape of 75.0 (length) × 4.0 (width) × 2 (thickness) mm$^3$. The tensile strength and elongation at break were attained from the stress–strain curve for the crosshead speed as 100 mm min$^{-1}$. The average values of five samples were reported.

The ideal energy performance of the CMDB propellant was calculated on the basis of the minimum free energy principle. The software Energy Calculation Star (v. 5.0) was designed in the Xi'an Modern Chemistry Research Institute [36]. The calculation conditions were set as follows. (1) The pressure inside the combustion chamber was 6.9 MPa and it was 0.1 MPa at ambient pressure; (2) nozzle expansion ratio was optimized, that is, the nozzle outlet pressure was equal to the ambient pressure; (3) nozzle outlet expansion angle equalled 0°; (4) the initial temperature was 298 K. Based on the equilibrium flow assumption, the combustion products of the composition immediately reached equilibrium during the expansion process.

The side walls of the 5 mm × 5 mm ×150 mm propellant strands were covered by polyvinyl alcohol and then dried with six repetitions. The burning rate of propellants was tested using a chimney-type strand burner system. At an initial temperature of 20°C and an AC voltage of 100 V, the propellant was ignited in the nitrogen-filled chamber using nickel–chromium alloy wire inserted in the top of the strand. Two low melting point fine fuses with a distance of 100 mm were inserted into the strand and the start and end times of the combustion signal were captured. The burning rate of propellant was calculated by five repeated experiments at each pressure. The standard deviation of the average experimental results obtained was 0.13–0.25. The burning rate pressure exponents of propellant labelled $n$ were obtained by Vieille's Law: $u = aP^n$.

The combustion wave structure of propellant strand of 7 mm × 7 mm × 120 mm was determined by a 25 µm diameter $\Pi$ type dual tungsten–rhenium thermocouple. The side walls of the propellant strands were covered with polyvinyl alcohol to achieve flame protection. The ignition was performed with nickel–chromium alloy wire. The combustion surface moved towards the thermocouple by degrees and finally reached the flame zone. Therefore, the entire combustion process of the condensed phase to the gaseous phase can be recorded.

# 3. Results and discussion

## 3.1. Radial distribution function of different atoms in NC/NG/GTA blend

The radial distribution function $g(r)$ is used to assess the possibility of another atom appearing at a distance $r$ from an assigned atom. The type of interaction forces could be determined from the position of the peaks and intensity could be determined from the heights of the peaks in the radial distribution function. Intermolecular action consists of hydrogen bonding and van der Waals (vdW) forces, which in turn include dipole–dipole, induction and dispersion interaction forces. Hydrogen bonding, strong vdW and weak vdW interaction forces correspond to distances of 2.6–3.1 Å, 3.1–5.0 Å or above 5.0 Å between atoms, respectively [18,35]. Herein, we explored the potential interaction force between hydroxyl functional group (–OH) in NC and nitrate functional group (–ONO$_2$) in NG

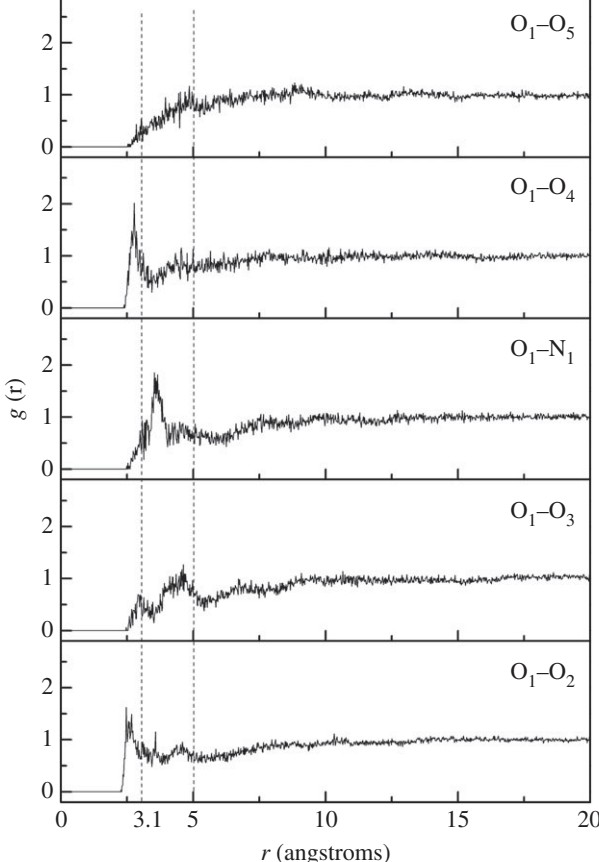

**Figure 3.** Radial distribution functions for different atoms of NC/NG/GTA blend.

plasticizer/acetate functional group ($-CH_3COO$) in GTA plasticizer. The radial distribution function curves of different atom pairs in NC polymer and plasticizers are shown in figure 3.

The $g(r)$ curves for the atom pairs $O_1-O_2$, $O_1-O_3$ formed between NC and NG and $O_1-O_4$ formed between NC and GTA in figure 3 showed peaks at distances of 2.61 Å, 2.92 Å and 2.73 Å respectively, verifying that a hydrogen bond interaction existed both between NC and NG and between NC and GTA. Besides, $O_1-O_4$ demonstrated a higher peak value of $g(r)$ by comparison with $O_1-O_2$ and $O_1-O_3$, suggesting that the hydrogen bond interaction between NC and GTA was stronger than that between NC and NG. The strong peaks of atom pairs $O_1-O_3$ and $O_1-N_1$ appeared in the range of 3.1 Å–5.0 Å, suggesting that the interaction force between atoms was a strong vdW force. The $g(r)$ curve for the atom pair $O_1-O_5$ of the NC/GTA had no peak, indicating there was no force between $O_1$ and $O_5$. It can be concluded that the major interaction force was hydrogen bond between NC and GTA, as well as the hydrogen bond and strong vdW forces between NC and NG.

## 3.2. Binding energy for NC and plasticizers

The binding energy ($E_{binding}$) can be used to evaluate the compatibility of two components in a blend. It is determined in terms of the negative value of the intermolecular interaction energy ($E_{inter}$): $E_{binding} = -E_{inter}$. The intermolecular interaction energy is calculated from the total energy of the blend and the energy of the individual components at equilibrium condition. The $E_{binding}$ of NC and plasticizer is calculated by the following formula.

$$E_{binding} = -E_{inter} = -(E_{total} - E_{NC} - E_{plasticizers}),$$ (3.1)

where $E_{total}$ is the total energy for NC/plasticizer blend. $E_{NC}$ and $E_{plasticizers}$ are the total energy for NC and plasticizers, respectively.

For comparison, the $E_{total}$, $E_{NC}$, $E_{plasticizers}$ and $E_{binding}$ of different NC/plasticizer blend are listed in table 2. The binding energy of NC/NG blend was greater than that of NC/NG/GTA blend, indicating that the compatibility of NC and NG was stronger than that of NC and GTA. This result further indicated

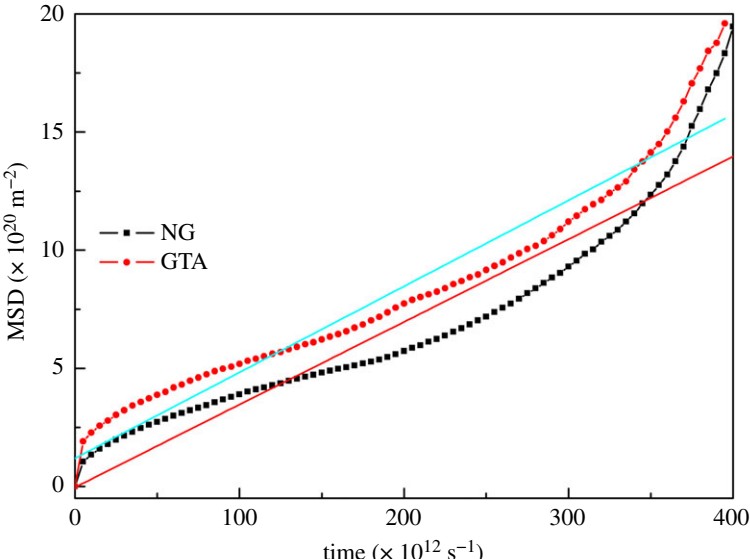

**Figure 4.** The MSD–$t$ curves and their linear fitted curves of plasticizer in NC matrix.

**Table 2.** Binding energy (kcal mol$^{-1}$) between NC and plasticizers.

| blend | $E_{total}$ | $E_{NC}$ | $E_{plasticizers}$ | $E_{binding}$ |
|---|---|---|---|---|
| NC/NG | −7466.01 | −1223.45 | −4999.76 | 1242.80 |
| NC/NG/GTA | −5859.16 | −839.74 | −4259.94 | 759.48 |

the conclusion that the intermolecular interaction between NC and NG is stronger than between NC and GTA.

## 3.3. Migration of plasticizer in NC matrix

The diffusion coefficient ($D$) is able to reflect the migration ability of a plasticizer, which can be determined by means of the MSD and simulation time ($t$) [37]. The formula for MSD of plasticizer is as follows:

$$\text{MSD} = S(t) = \left\langle |r_t - r_0|^2 \right\rangle, \tag{3.2}$$

where $r_0$ and $r_t$ are the locations of the plasticizer at time 0 and $t$, respectively. Based on the Einstein equation, the expression of $D$ is obtained as

$$D = \lim_{t \to \infty} \frac{\left\langle |r_t - r_0|^2 \right\rangle}{6t}. \tag{3.3}$$

The ultimate $D$ value is calculated by combination of equations (3.2) and (3.3):

$$D = \frac{S(t)}{6t} = \frac{a}{6}, \tag{3.4}$$

where $a$ represents the slope of MSD–$t$ curve.

 With the aim of further investigating the influence of GTA and NG plasticizers on the plasticization ability of NC, the MSD–$t$ and the linear fitting curves reflecting the migration ability of GTA and NG plasticizers into the NC matrix are shown in figure 4. It is thus clear that the MSD of GTA and NG plasticizers in NC matrix was almost linearly dependent on the simulation time. GTA always displayed a higher MSD value than NG, indicating that the migration ability of GTA in NC matrix was higher than that of NG. In the end, the diffusion coefficients ($D$) of GTA and NG plasticizers in NC binder calculated by equation (3.4) were $6.07 \times 10^{-11}$ m$^2$ s$^{-1}$ and $5.83 \times 10^{-11}$ m$^2$ s$^{-1}$, respectively.

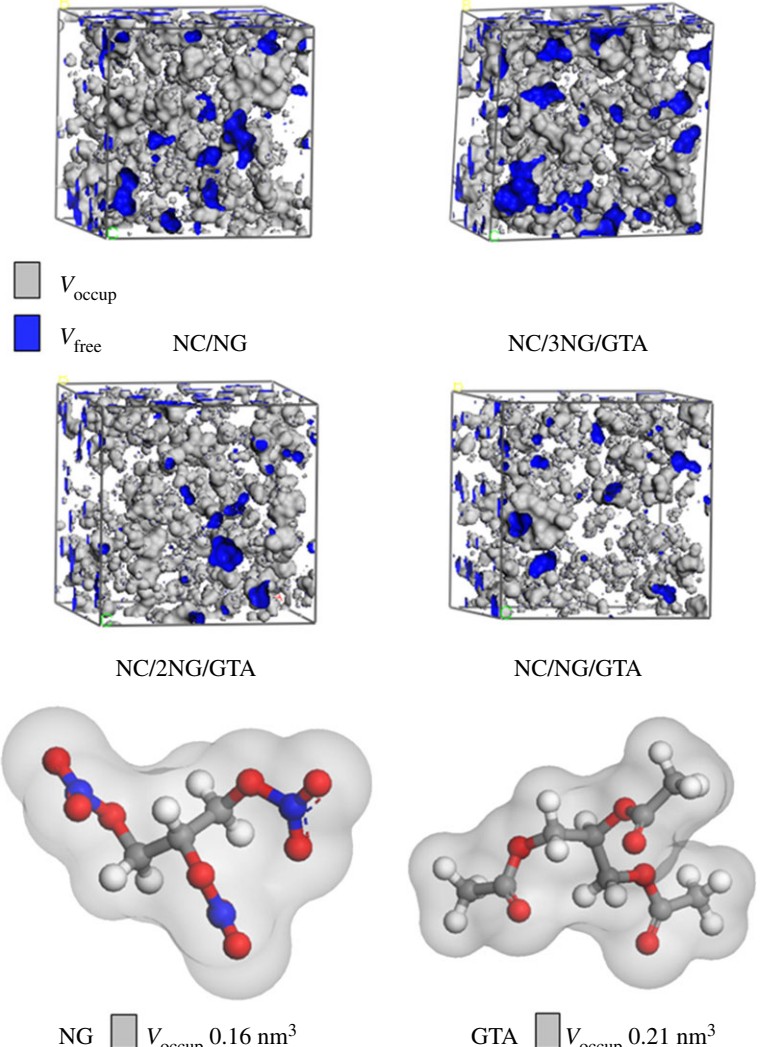

NG $V_{\text{occup}}$ 0.16 nm$^3$        GTA $V_{\text{occup}}$ 0.21 nm$^3$

**Figure 5.** Volume distribution of NC/plasticizers blends and volume of plasticizer molecules.

This illustrated that GTA was easier to migrate in NC. This result also verified that the plasticization efficiency of GTA on NC binder was stronger than that of NG.

## 3.4. Volume distribution of NC/plasticizers blend

The volume distribution profile of NC/plasticizers blend was determined by MD simulation analysis. In figure 5, the grey region is the occupied volume of NC/plasticizers blend ($V_{\text{occup}}$), and the blue region is the free volume ($V_{\text{free}}$), indicating the intermolecular interstitial space. Table 3 lists the data of $V_{\text{occup}}$ and $V_{\text{free}}$ for NC/plasticizers blends. The simulated $V_{\text{occup}}$ was larger for different NC/NG/GTA blends than for the NC/NG blend. Simultaneously, the volumes for individual GTA and NG molecules calculated in figure 5 were 0.21 nm$^3$ and 0.16 nm$^3$, consistent with the fact that the $V_{\text{occup}}$ value of NC/NG/GTA was higher than those of NC/NG. Especially, $V_{\text{free}}$ value of NC/3NG/GTA blend was significantly greater than those of NC/NG blend, which was beneficial to the movement of molecular chains and brought about an increase of structural flexibility. Besides, $V_{\text{free}}$ values of NC/NG/GTA blends were found to decrease with increasing GTA content in mixed plasticizers, which increased the rotational resistance of NC molecular chains and thus led to the decrease in mechanical property.

## 3.5. Mechanical property of NC/plasticizers blend

The mechanical property of propellant is very important, which affects its processability and safety. The relationship of stress and strain for material abides by the universal Hooke's Law, and

**Table 3.** Occupied and free volume of NC/plasticizers blends.

| models | NC/NG | NC/3NG/GTA | NC/2NG/GTA | NC/NG/GTA |
|---|---|---|---|---|
| $V_{occup}$ (nm$^3$) | 36.61 | 37.77 | 38.67 | 39.30 |
| $V_{free}$ (nm$^3$) | 6.68 | 7.62 | 4.33 | 3.56 |

**Table 4.** The elastic coefficients and moduli of NC/plasticizers blends (in GPa).

| constants | NC/NG | NC/3NG/GTA | NC/2NG/GTA | NC/NG/GTA |
|---|---|---|---|---|
| $C_{11}$ | 13.92 | 12.13 | 11.73 | 12.79 |
| $C_{22}$ | 13.89 | 15.77 | 13.33 | 13.34 |
| $C_{33}$ | 14.01 | 10.22 | 12.70 | 12.52 |
| $C_{44}$ | 4.50 | 2.20 | 3.61 | 4.03 |
| $C_{55}$ | 3.77 | 2.27 | 3.73 | 4.15 |
| $C_{66}$ | 4.66 | 3.04 | 2.55 | 3.60 |
| $C_{12}$ | 6.97 | 7.11 | 6.88 | 6.33 |
| $C_{13}$ | 6.46 | 5.66 | 6.49 | 6.13 |
| $C_{23}$ | 6.76 | 5.65 | 6.06 | 6.75 |
| $E$ | 11 | 6.89 | 8.73 | 10.07 |
| $K$ | 5.47 | 7.97 | 6.19 | 5.21 |
| $G$ | 4.31 | 2.50 | 3.30 | 3.93 |
| $C_{12}-C_{44}$ | 2.47 | 4.91 | 3.27 | 2.3 |
| $\gamma$ | 0.28 | 0.38 | 0.32 | 0.28 |
| $K/G$ | 1.27 | 3.19 | 1.88 | 1.33 |

the equation is

$$\sigma_i = C_{ij}\varepsilon_j, \tag{3.5}$$

where $C_{ij}$ ($i,j = 1$–$6$) is a ($6 \times 6$) matrix of elastic coefficient. By considering the strain energy, the matrix of elastic coefficient is symmetrical, that is, $C_{ij} = C_{ji}$. Consequently, it has 21 elastic coefficients to express the relationship between stress and strain of anisotropic material. With regard to most commonly used anisotropic materials, the 21 elastic coefficients $C_{ij}$ are independent of each to other. An isotropic material has only two independent elastic coefficients, which are called Lame coefficients ($\lambda$ and $\mu$). Using $\lambda$ and $\mu$, the tensile modulus $E$, shear modulus $G$, bulk modulus $K$ and Poisson's ratio $\gamma$ are calculated by the following equations [38]. The program supposes that the material is isotropic, and the computational isotropic mechanical properties of the material are obtained.

and

$$\left. \begin{aligned} E &= \frac{\mu(3\lambda + 2\mu)}{\lambda + \mu}, \\ K &= \lambda + \frac{2}{3\mu}, \\ \gamma &= \frac{\lambda}{2(\lambda + \mu)} \\ G &= \mu. \end{aligned} \right\} \tag{3.6}$$

Poisson's ratio ($\gamma$) is often used to assess the plasticity of materials. There is a relation between Poisson's ratio and different moduli:

$$E = 3K(1 - 2\gamma) = 2G(1 + \gamma). \tag{3.7}$$

Due to the matrix of elastic coefficients being symmetric, table 4 only lists partial coefficients of different blends. According to the difference of the elastic coefficients of different blends, it can be

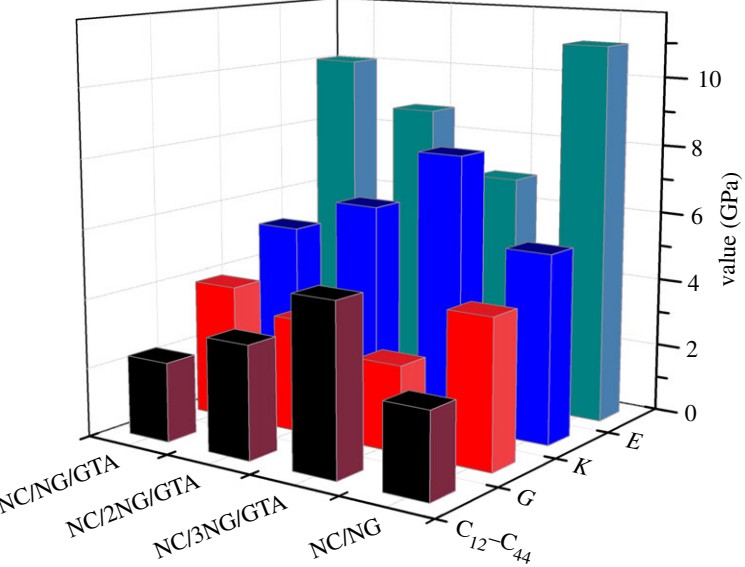

**Figure 6.** Variation of elastic moduli and Cauchy pressure for NC/plasticizers blends.

found that the blends have anisotropic behaviour to some extent. Tensile modulus ($E$), bulk modulus ($K$) and shear modulus ($G$) are applied to judge the capacity of a material to undergo elastic deformation. To observe the changes more obviously, figure 6 was plotted. From table 4, most elastic coefficients and moduli of NC/NG/GTA blends were found to decrease compared with the NC/NG blend, indicating that the elasticity of the NC/NG/GTA blends increased. It indicated that the synergistic interaction of NG/GTA mixed plasticizers reduced the stiffness and improved the flexibility of NC. In addition, most elastic coefficients and moduli of NC/NG/GTA blends were found to increase with the increased GTA content in mixed plasticizers, predicted that the elasticity and plasticity of blends decreased. This might be due to the smaller polarity and larger volume of GTA than NG, which led to a reduction of intermolecular interaction among the NC and plasticizers. When the GTA substituted for NG, the slip and bending of NC molecular chain became difficult under external forces, leading to an increase in mechanical strength.

Cauchy pressure ($C_{12}$–$C_{44}$) is a measure of ductility and brittleness for materials, with positive values indicating that the material is ductile and negative values indicating that the material is brittle. The data of table 4 predicted that compared with NC/NG blend, ($C_{12}$–$C_{44}$) value of the NC/NG/GTA blend was increased with a different extent. And the ($C_{12}$–$C_{44}$) values of the NC/NG/GTA blends decreased with the increased GTA content in mixed plasticizers, which implied the ductility of blends decreased. Toughness is the ability of a material to deform after absorbing energy, which can be determined by the ratio $K/G$. The higher the $K/G$ value, the better the toughness of the material. It is thus clear in table 4 that the changes of $K/G$ and ($C_{12}$–$C_{44}$) were consistent, illustrating that the toughness of NC/3NG/GTA blend was the best. The Poisson's ratio of different blends was between 0.28 and 0.38, showing the better plasticity. To summarize, NG/GTA mixed plasticizers increased the elasticity, ductility and tenacity of the NC.

## 3.6. Mechanical properties of propellant based on NG/GTA mixed plasticizers

Mechanical properties of propellant are closely related to structural integrity and performance stability in practical application. The mechanical properties of nitramine propellant are not only related to binder matrix but also to the interactions between binder matrix and additives. The mechanical properties of propellant based on NG/GTA mixed plasticizers are shown in figure 7 at −40°C, 20°C and 50°C.

It can be seen that propellants displayed brittle failure, that is, higher tensile strength and lower elongation at break at low temperatures. This phenomenon was mainly owing to the freeze of the molecular chains of NC, which greatly restricted the movement of the chains. However, at elevated temperature the propellant demonstrated ductility characteristics, i.e. higher elongation at break and lower tensile strength. Especially, the elongation at break of the propellant increased greatly with an increase of temperature, while the tensile strength decreased significantly. This is due to the softening

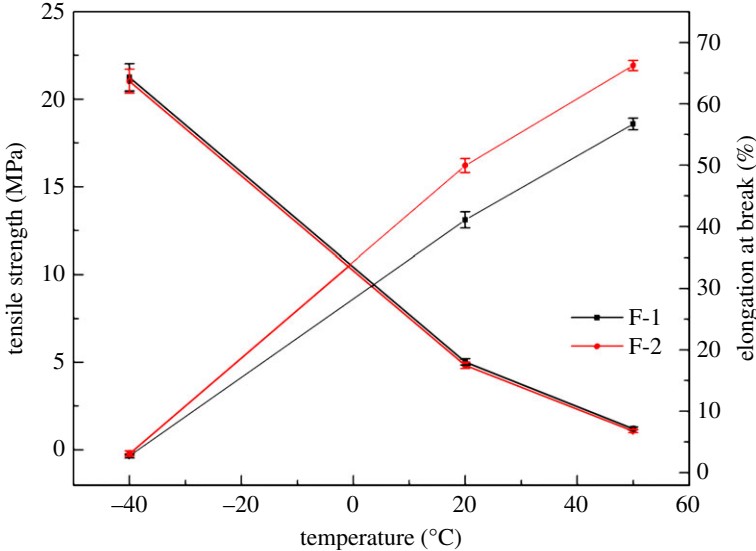

**Figure 7.** The tensile strength and elongation at break of propellants at different temperatures.

of the NC binder matrix at high temperature, which led to the weakening of entanglement ability between molecule chains and the increase of ductility. Moreover, the elongation at break of the propellant based on NG/GTA mixed plasticizers was significantly higher than that of the propellant with NG plasticizer at the same temperature. Meanwhile, the tensile strength of the propellant based on NG/GTA mixed plasticizers was only slightly reduced. These results were consistent with the MD simulation results. From the simulation of NC/plasticizers blend, the mechanical properties of propellant could be enhanced by mixed plasticizers owing to the large $V_{free}$ and lower $C_{ij}$, which could enhance the movement ability of the NC molecular chains. It could be inferred that the mechanical properties might be the macroscopic reaction of the plasticization efficiency of NG/GTA mixed plasticizers on NC binder.

## 3.7. Energetic properties of propellants based on NG/GTA mixed plasticizers

Energy properties are an important factor in assessing propellant performance. Propellant decomposes into elevated temperature working fluids in a combustion chamber, which is a process of thermal and chemical equilibrium (constant of energy and mass). Based on the isenthalpic principle, thermodynamic parameters such as combustion chamber temperature ($T_c$) and equilibrium product components in the combustion chamber could be directly obtained. By means of the isentropic expansion of combustion product in the nozzle, we can obtain the outlet species composition, outlet temperature ($T_e$), etc. The standard theoretical specific impulse ($I_{sp}$) is the major parameter to evaluate the energy efficiency of propellant, which has a remarkable impact on missile range. The larger the $I_{sp}$, the further the range of a missile.

We have designed some formulations to study the effect of NG/GTA mixed plasticizers on the energetic characteristics of propellant. The theoretical calculation results for energetic parameters of propellant at 6.9 MPa are summarized in table 5. It was thus clear that these energetic parameters such as $I_{sp}$, $C_v$, $T_c$, $T_e$, oxygen coefficients, relative molecular mass of gaseous product, density and heat of explosion of propellants containing NG/GTA mixed plasticizers were lower than those of propellants containing NG plasticizer under an identical mass content of plasticizer. Meanwhile, with the increased non-energetic plasticizer GTA content in mixed plasticizers, the energetic characteristic values of propellants were decreased significantly. Therefore, the energy, combustion and mechanical properties should be considered comprehensively in the formulation design of high-performance propellant.

## 3.8. Combustion property of propellant based on NG/GTA mixed plasticizers

The burning rate of the propellant influences the gas generation speed, which in turn fixes the internal pressure and total thrust of the motor. The strand burning rate tests were carried out at 2–22 MPa to

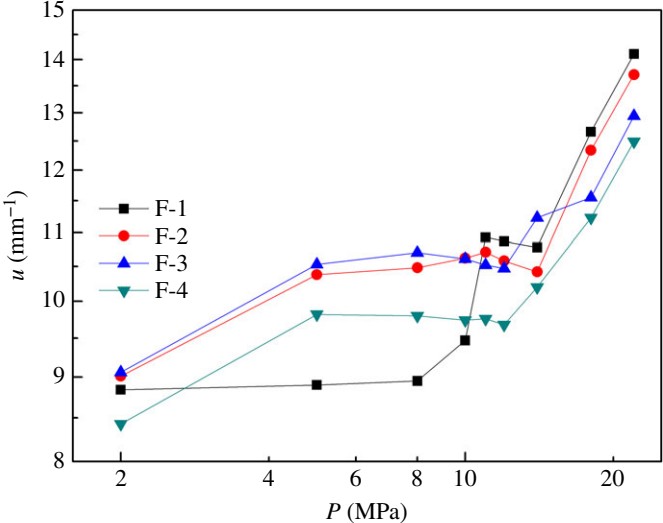

**Figure 8.** Burning rate of propellant based on NG/GTA at different pressure.

**Table 5.** The energetic characteristics[a] of propellants based on NG/GTA mixed plasticizers.

| no. | $I_{sp}$ (N s kg$^{-1}$) | $C_v$ (m s$^{-1}$) | $\varphi$ | $M_c$ | $T_e$ (K) | $T_c$ (K) | $\rho$ (g cm$^{-3}$) | $Q$ (cal g$^{-1}$) |
|---|---|---|---|---|---|---|---|---|
| F-1 | 2380.22 | 1498.51 | 0.65 | 25.01 | 1338.33 | 2898.32 | 1.75 | 983.83 |
| F-2 | 2202.45 | 1401.22 | 0.57 | 23.30 | 1033.94 | 2389.60 | 1.71 | 797.71 |
| F-3 | 2136.30 | 1362.59 | 0.54 | 22.75 | 941.91 | 2213.38 | 1.70 | 730.67 |
| F-4 | 1988.52 | 1273.81 | 0.50 | 21.70 | 767.68 | 1856.50 | 1.68 | 595.95 |

[a]$I_{sp}$ is ideal specific impulse, $C_v$ is characteristic velocity, $\varphi$ is oxygen coefficient, $M_c$ is relative molecular mass of gaseous products, $T_e$ is outlet temperature, $T_c$ is adiabatic flame temperature, $\rho$ is density and $Q$ is heat of explosion.

study the effect of NG/GTA mixed plasticizers on combustion performance of propellants. It is thus clear in figure 8 that the burning rates of the four specimens increased to varying degrees with the increase of pressure. The burning rate of propellant containing NG/GTA mixed plasticizers was higher than that of propellant containing NG plasticizer at 2–11 MPa, indicating that the combustion performance of propellants containing NG/GTA mixed plasticizers was superior to that of propellants containing NG plasticizer. This was further verified by MD simulation that the NG/GTA mixed plasticizers had better ability to plasticize NC. From the viewpoint of heat transfer, the increase of heat release and feedback to the combustion surface during combustion was the primary factor. However, the combustion properties of propellants at 12–22 MPa were contrary to those of propellants at 2–11 MPa. From the viewpoint of energy, the addition of non-energetic plasticizer GTA to the propellant formulation reduced the energy during the combustion, which was the reason for the decreased burning rate at high-pressure region. The burning rate pressure exponent of propellants was calculated in the pressure range of 2–22 MPa and is displayed in table 6. Propellants had better combustion platform at 2–11 MPa, and combustion platform of propellant containing NG/GTA mixed plasticizers moved to medium-pressure zone. It is worth noting that the burning rate pressure exponent increased when the pressure was greater than 12 MPa. The burning rate of propellants is affected by various factors, such as the plasticity and uniformity of the blends, plasticizer types, pressure, temperature etc. [39,40]. This is probably the reason for the elevated burning rate pressure exponents of propellant in high-pressure zone.

## 3.9. Combustion wave structure of propellant based on NG/GTA mixed plasticizers

The energy generated during the combustion of a traditional CMDB propellant is transferred from the bright flame area to the dark area, and then to the propellant surface [41]. The combustion wave structure of the propellant with 3NG/GTA mixed plasticizers was determined at different pressures

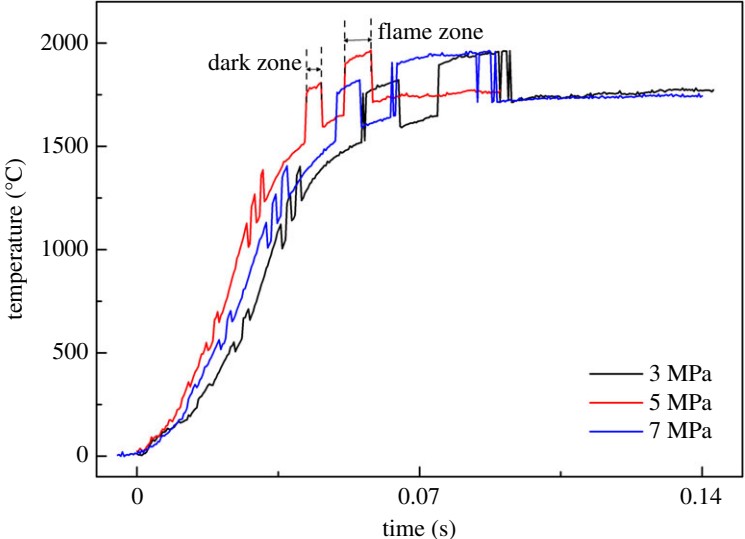

**Figure 9.** Temperature profiles in the combustion waves of the propellant with 3NG/GTA mixed plasticizers.

**Table 6.** Burning rate pressure exponents of propellants based on NG/GTA.

| no. | burning rate pressure exponents | | | |
|-----|-----------|-----------|-----------|-----------|
|     | 2–5 MPa | 5–12 MPa | 12–22 MPa | 2–22 MPa |
| F-1 | 0.007 | 0.235 | 0.466 | 0.187 |
| F-2 | 0.154 | 0.030 | 0.469 | 0.137 |
| F-3 | 0.164 | −0.005 | 0.316 | 0.116 |
| F-4 | 0.167 | −0.014 | 0.417 | 0.128 |

and is exhibited in figure 9. It was noted that at different pressures the temperature first increased quickly, then in dark zone the temperature rise was slowed down. The temperature-rise rate at different pressures was 5 MPa > 7 MPa > 3 MPa. This was owing to the improvement of thermal feedback from gaseous phase to the dark area and then to solid phase in the combustion process. This was fitted well with a negative burning rate pressure exponent of propellant with NG/GTA mixed plasticizers in the medium-pressure zone. More interestingly, although the temperature-rise rate at different pressures was different, the combustion temperature in luminous flame zones was almost identical. This phenomenon during the combustion of propellants suggested that the combustion temperature in gaseous phases was less sensitive to pressure, that is, the combustion temperature was almost invariant with the increase of pressure. To sum up, the combustion wave structure further confirmed the above proposition that NG/GTA mixed plasticizers could enhance the combustion efficiency of propellants effectively at low pressures, facilitating the heat feedback to propellant.

## 4. Conclusion

(1) The radial distribution function indicated that NC/plasticizers blends showed stronger intermolecular interaction of van der Waals force and hydrogen bonds. The binding energy of NC/NG blends was 1242.80 kcal mol$^{-1}$, higher than that of NC/NG/GTA blends (759.48 kcal mol$^{-1}$).

(2) The MSDs of plasticizers revealed that the diffusion coefficient ($D$) of GTA plasticizer in the NC binder was $6.07 \times 10^{-11}$ m$^2$ s$^{-1}$, higher than that of NG in the NC binder ($5.83 \times 10^{-11}$ m$^2$ s$^{-1}$). The free volume $V_{\text{free}}$ of NC/3NG/GTA blend was greater than that of NC/NG blend, which was beneficial to the structural ductility.

(3) Mechanical properties determined from the MD simulation confirmed that the NG/GTA mixed plasticizers increased the elasticity, ductility and tenacity of the NC. Meanwhile, the mechanical properties tests displayed that the elongation at break of the propellant based on NG/GTA mixed plasticizers was significantly higher than that of the propellant with NG plasticizer at the same temperature.

(4) The burning rates of propellants containing NG/GTA mixed plasticizers were higher than those of propellants containing NG plasticizer at low pressure. The combustion platform of propellant containing NG/GTA mixed plasticizers moved to medium-pressure zone. The combustion wave temperature demonstrated that the combustion temperature in gaseous phases was less affected by pressure.

In general, these experimental results coincide with the MD simulation. NG/GTA mixed plasticizers had better ability to plasticize NC, could enhance the combustion efficiency of propellants effectively at low pressures, and facilitated the heat feedback to propellant. The energy, combustion and mechanical properties should be considered comprehensively in the formulation design of high-performance propellant.

Data accessibility. Our data are deposited at Dryad: https://doi.org/10.5061/dryad.bk3j9kdbz for review [42] and https://doi.org/10.5061/dryad.tx95x69xk for publication [43].

Authors' contributions. Conceptualization, L.Y. and X.W.; methodology, X.W. and J.L.; software, T.C.; investigation, T.C.; data curation, M.L.; data analysis, Q.H.; writing—review and editing, L.Y. All authors discussed the results and contributed to the final manuscript. All authors have read and agreed to the published version of the manuscript.

Competing interests. The authors declare no competing interests.
Funding. This research was supported by the Key Laboratory Foundation for Advance Research of China (grant no. 6142603190306).

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
