## [Peer Review File · Royal Society Open Science]

Review History

RSOS-210576.R0 (Original submission)

Review form: Reviewer 1

Is the manuscript scientifically sound in its present form?

Yes

Are the interpretations and conclusions justified by the results?

Yes

Is the language acceptable?

No

Do you have any ethical concerns with this paper?

No

Have you any concerns about statistical analyses in this paper?

No

Recommendation?

Accept with minor revision (please list in comments)

Comments to the Author(s)

Authors of this manuscript entitled "Study on Structure and Property of Propellant based on Nitroglycerine/Glycerol Triacetate Mixed Plasticizers" presented a systematic investigation through MD simulations to investigate the influence of NG/GTA mixed plasticizers on the plasticizing ability of NC binder in solid propellant. The intermolecular interaction between NC and plasticizers has been evaluated using the radial distribution function and the binding energy. They have performed a comprehensive literature survey to carry out the present work. Further, they conducted experimental studies on the effects of NG/GTA mixed plasticizers on the energy and burning performance of CMDDB propellant. The results obtained from this study can offer direction for the designs of high energy materials and provide guidance for the applications of NG/GTA mixed plasticizers in high performance propellant. Therefore, it is worthwhile to publish in Royal Society Open Science.

Few comments to the authors:

- 1) Authors need to correct title of the manuscript.
- 2) In abstract: Please reframe the sentence...." Meanwhile, the mechanical properties of the propellant....."
- 3) Authors need to refine the English language throughout the manuscript.
- 4) How authors can abandon to refer the most relevant recent work on binder and plasticizers (10.1080/10601325.2019.1669458 and 10.1039/c5ra16476a) ?
- 5) Precise conclusion need to draw by the authors.

Review form: Reviewer 2

Is the manuscript scientifically sound in its present form?

No

Are the interpretations and conclusions justified by the results?

Yes

Is the language acceptable?

Yes

Do you have any ethical concerns with this paper?

No

Have you any concerns about statistical analyses in this paper?

No

Recommendation?

Reject

Comments to the Author(s)

The influence of nitroglycerine (NG)/glycerol triacetate (GTA) mixed plasticizers on the plasticizing ability of nitrocellulose (NC) binder was studied by Molecular dynamics (MD) simulation.

no novelty and interestingly results are obtained. Van der Waals and hydrogen bonds between NC and GTA has been been recognized. moreover, The energy and combustion properties of

propellants were further investigated. However, addition NG could enhance energy and combustion due to energetic group (-NO₂).

- 1 the molecular structure should be given in Fig 1
- 2 what mean of F-1 F-2 and F-3 in table 1.
- 3 no clear experimental process of combustion
- 4 no clear experimental process of mechanical properties

Review form: Reviewer 3

Is the manuscript scientifically sound in its present form?

Yes

Are the interpretations and conclusions justified by the results?

Yes

Is the language acceptable?

No

Do you have any ethical concerns with this paper?

No

Have you any concerns about statistical analyses in this paper?

No

Recommendation?

Major revision is needed (please make suggestions in comments)

Comments to the Author(s)

In this work, authors used molecular dynamics simulation to investigate the influence of nitroglycerine (NG)/glycerol triacetate (GTA) mixed plasticizers on the nitrocellulose (NC) binder in solid propellant. This work carried out molecular dynamics simulation studies on the important properties of propellants and plasticizers, such as mechanical properties, combustion properties, etc. However, in my opinion, this manuscript cannot be accepted in current form. Here below is my suggestion.

1. The English of the manuscript needs major revision, and many sentences cannot accurately express the meaning. In particular, the topic of the paper needs to be corrected, and the current topic is prone to ambiguity.
2. Authors has done a lot of molecular dynamics simulations, but there are not enough actual experimental results to corroborate the simulation results. Such as combustion and mechanical properties.
3. According to the description in the manuscript, mixed plasticizers can help improve the properties of the propellant. But there is no comparative experiment. There is no specific experimental data to prove the significance of adding triacetin. A comparison can lead to more convincing conclusions.
4. In the statement of the conclusion part, specific experimental data is needed to support the author's conclusion.

Decision letter (RSOS-210576.R0)

Dear Mrs Yang:

Manuscript ID: RSOS-210576

Title: "Study on Structure and Property of Propellant based on Nitroglycerine/Glycerol Triacetate Mixed Plasticizers"

Thank you for submitting the above manuscript to Royal Society Open Science. Your paper was sent to reviewers and their comments are included at the bottom of this letter.

In view of the concerns raised by the reviewers, the manuscript has been rejected in its current form. However, a new manuscript may be submitted which takes into consideration these comments.

Please note that resubmitting your manuscript does not guarantee eventual acceptance, and that your resubmission will be subject to peer review before a decision is made.

Your resubmitted manuscript should be submitted by 29-Nov-2021. If you are unable to submit by this date please contact the Editorial Office.

On behalf of the Subject Editor Professor Anthony Stace and the Associate Editor Professor Chaohua Cui

REVIEWER(S) REPORTS:
Associate Editor Comments to Author ():
RSC Associate Editor:

Comments to the Author:
(There are no comments.)

RSC Subject Editor:
Comments to the Author:
(There are no comments.)

Reviewers' Comments to Author:

Reviewer: 1

Comments to the Author(s)

Authors of this manuscript entitled "Study on Structure and Property of Propellant based on Nitroglycerine/Glycerol Triacetate Mixed Plasticizers" presented a systematic investigation through MD simulations to investigate the influence of NG/GTA mixed plasticizers on the plasticizing ability of NC binder in solid propellant. The intermolecular interaction between NC and plasticizers has been evaluated using the radial distribution function and the binding energy. They have performed a comprehensive literature survey to carry out the present work. Further, they conducted experimental studies on the effects of NG/GTA mixed plasticizers on the energy and burning performance of CMDDB propellant. The results obtained from this study can offer direction for the designs of high energy materials and provide guidance for the applications of NG/GTA mixed plasticizers in high performance propellant. Therefore, it is worthwhile to publish in Royal Society Open Science.

Few comments to the authors:

- 1) Authors need to correct title of the manuscript.
- 2) In abstract: Please reframe the sentence...." Meanwhile, the mechanical properties of the propellant....."
- 3) Authors need to refine the English language throughout the manuscript.
- 4) How authors can abandon to refer the most relevant recent work on binder and plasticizers (10.1080/10601325.2019.1669458 and 10.1039/c5ra16476a) ?
- 5) Precise conclusion need to draw by the authors.

Reviewer: 2

Comments to the Author(s)

The influence of nitroglycerine (NG)/glycerol triacetate (GTA) mixed plasticizers on the plasticizing ability of nitrocellulose (NC) binder was studied by Molecular dynamics (MD) simulation. no novelty and interestingly results are obtained. Van der Waals and hydrogen bonds between NC and GTA has been been recognized. moreover, The energy and combustion properties of propellants were further investigated. However, addition NG could enhance energy and combustion due to energetic group (-NO₂).

- 1 the molecular structure should be given in Fig 1
- 2 what mean of F-1 F-2 and F-3 in table 1.
- 3 no clear experimental process of combustion
- 4 no clear experimental process of mechanical properties

Reviewer: 3

Comments to the Author(s)

In this work, authors used molecular dynamics simulation to investigate the influence of nitroglycerine (NG)/glycerol triacetate (GTA) mixed plasticizers on the nitrocellulose (NC) binder in solid propellant. This work carried out molecular dynamics simulation studies on the important properties of propellants and plasticizers, such as mechanical properties, combustion

properties, etc. However, in my opinion, this manuscript cannot be accepted in current form. Here below is my suggestion.

1. The English of the manuscript needs major revision, and many sentences cannot accurately express the meaning. In particular, the topic of the paper needs to be corrected, and the current topic is prone to ambiguity.
2. Authors has done a lot of molecular dynamics simulations, but there are not enough actual experimental results to corroborate the simulation results. Such as combustion and mechanical properties.
3. According to the description in the manuscript, mixed plasticizers can help improve the properties of the propellant. But there is no comparative experiment. There is no specific experimental data to prove the significance of adding triacetin. A comparison can lead to more convincing conclusions.
4. In the statement of the conclusion part, specific experimental data is needed to support the author's conclusion.

Author's Response to Decision Letter for (RSOS-210576.R0)

See Appendix A.

RSOS-211033.R0

Review form: Reviewer 1

Is the manuscript scientifically sound in its present form?

Yes

Are the interpretations and conclusions justified by the results?

Yes

Is the language acceptable?

Yes

Do you have any ethical concerns with this paper?

No

Have you any concerns about statistical analyses in this paper?

No

Recommendation?

Accept as is

Comments to the Author(s)

Authors of this manuscript have revised it as per suggestions/comments. Therefore, I recommend for its publication in Royal Society Open Science.

Review form: Reviewer 4

Is the manuscript scientifically sound in its present form?

Yes

Are the interpretations and conclusions justified by the results?

Yes

Is the language acceptable?

Yes

Do you have any ethical concerns with this paper?

No

Have you any concerns about statistical analyses in this paper?

Yes

Recommendation?

Accept as is

Comments to the Author(s)

This is a revised manuscript. I read it carefully and the authors revised according to Reviewers' comments. I think it can be accepted for publication.

Decision letter (RSOS-211033.R0)

Dear Mrs Yang:

Title: Structure and Property of Propellant based on Nitroglycerine/Glycerol Triacetate Mixed Plasticizers: MD Simulation and Experimental Study
Manuscript ID: RSOS-211033

It is a pleasure to accept your manuscript in its current form for publication in Royal Society Open Science. The chemistry content of Royal Society Open Science is published in collaboration with the Royal Society of Chemistry.

Yours sincerely,
Dr Ellis Wilde
Publishing Editor, Journals

On behalf of the Subject Editor Professor Anthony Stace and the Associate Editor Professor Chaohua Cui.

RSC Associate Editor
Comments to the Author:
(There are no comments.)

Reviewer(s)' Comments to Author:
Reviewer: 1

Comments to the Author(s)
Authors of this manuscript have revised it as per suggestions/comments. Therefore, I recommend for its publication in Royal Society Open Science.

Reviewer: 4

Comments to the Author(s)
This is a revised manuscript. I read it carefully and the authors revised according to Reviewers' comments. I think it can be accepted for publication.

Appendix A

Dear editor,

First of all, we appreciate the constructive comments from your respected editorial journal team and reviewers concerning our manuscript. We have revised the whole manuscript carefully according to the reviewer comments and tried to avoid any grammar or syntax error. Following modifications have been made in the revised version of this manuscript. And detailed corrections are as follows.

Comments to the Author

Reviewer: 1

1) Authors need to correct title of the manuscript.

Response:

After careful consideration, we have revised the title of the manuscript. The new title is as follows.

Structure and Property of Propellant based on Nitroglycerine/Glycerol Triacetate Mixed Plasticizers: MD Simulation and Experimental Study

2) In abstract: Please reframe the sentence...." Meanwhile, the mechanical properties of the propellant....."

Correction:

Then, the mechanical properties of the propellant based on NG/GTA mixed plasticizers were investigated systematically using experimental and simulation calculation method.

3) Authors need to refine the English language throughout the manuscript.

Response:

We have revised the whole manuscript carefully and tried to avoid any grammar or syntax error.

4) How authors can abandon to refer the most relevant recent work on binder and plasticizers (10.1080/10601325.2019.1669458 and 10.1039/c5ra16476a) ?

Response:

We have studied these two relevant recent works on binder and plasticizers carefully and cited them in our manuscript. See references [21] and [33].

21 Vijayalakshmi, R., Agawane, N. T., Talawar, M. B. and Khan, M. 2019 Examining the compatibility of energetic plasticizer DNDA-5 with energetic binders. *J. Macromol. Sci. A*, **57(1)**, 1-9.

33 Kumarshee, S., Reddy, S. T., JavaidAthar, Kantisikder, A., Talawar, M. B. and ShaibalBanerjee, et al. 2015 Probing the compatibility of energetic binder poly-glycidyl nitrate with energetic plasticizers: thermal, rheological and DFT studies. *RSC Adv.*, **5 (123)**, 101297-101308.

5) Precise conclusion need to draw by the authors.

Response:

We have revised the conclusion of the manuscript as follows.

(1) The radial distribution function indicated that NC/plasticizers blends showed stronger intermolecular interaction of van der Waals force and hydrogen bonds. The binding energy of NC/NG blends was 1242.80 kcal/mol, higher than that of NC/NG/GTA blends (759.48 kcal/mol).

(2) The mean-squared displacements of plasticizers revealed that the diffusion coefficients (D) of GTA plasticizer in the NC binder was 6.07×10^{-11} m²/s, higher than that of NG in the NC binder (5.83×10^{-11} m²/s). The free volume V_{free} of NC/3NG/GTA blend was greater than that of NC/NG blend, which was beneficial to the structural ductility.

(3) Mechanical properties determined from the MD simulation confirmed that the NG/GTA mixed plasticizers increased the elasticity, ductility, and tenacity of the NC. Meanwhile, the mechanical properties tests displayed that the elongation at break of the propellant based on NG/GTA mixed plasticizers was significantly higher than that of the propellant with NG plasticizer at the same temperature.

(4) The burning rates of propellants containing NG/GTA mixed plasticizers were higher than of propellants containing NG plasticizer at low pressure. The combustion platform of propellant containing NG/GTA mixed plasticizers moved to medium pressure zone. The combustion wave temperature demonstrated that the combustion temperature in gaseous phases was less affected by pressure.

In general, these experimental results coincide with the MD simulation. NG/GTA mixed plasticizers had better ability to plasticize nitrocellulose, could enhance the combustion efficiency of propellants effectively at low pressures, facilitated the heat feedback to propellant. The energy, combustion and mechanical properties should be considered comprehensively in the formulation design of high-performance propellant.

Reviewer: 2

1 the molecular structure should be given in Fig 1

Response:

We add the molecular structure to Figure 1 and modify it as follows.

Fig. 1. Molecular (a) and optimized chemical structures (b) of the NC polymer, NG and GTA plasticizers

2 what mean of F-1, F-2 and F-3 in table 1.

Response:

F-1, F-2 and F-3 denote composition number respectively.

3 no clear experimental process of combustion

Response:

We have revised the manuscript to describe the combustion performance test experimental process in more detail. The details are as follows.

The combustion wave structure of propellant strand as 7 mm×7 mm×120 mm was determined by a $\Phi = 25 \mu\text{m}$ Π type dual tungsten-rhenium thermocouple. The side walls of the propellant strands were covered with polyvinyl alcohol to achieve flame protection. The ignition was performed with nickel-chromium alloy wire. The combustion surface moved towards the thermocouple by degrees and finally reached the flame zone. Therefore, the entire combustion process of the condensed phase to the gaseous phase can be recorded.

4 no clear experimental process of mechanical properties

Response:

We have revised the manuscript to describe the mechanical properties test experimental process in more detail. The details are as follows.

Tensile test was carried out on an Instron 4505 (USA) universal testing machine with a temperature of -40°C , 20°C and 50°C , respectively. The cured propellant sample was cut into slices with thickness 2mm, and the slice was stamped into a dumbbell shape of 75.0(length) *4.0 (width) *2 (thickness) mm^3 . The tensile strength and elongation at break were attained from the stress-strain curve for the crosshead speed as $100\text{mm}\cdot\text{min}^{-1}$. The average values of five samples were reported.

Reviewer: 3

1. The English of the manuscript needs major revision, and many sentences cannot accurately express the meaning. In particular, the topic of the paper needs to be corrected, and the current topic is prone to ambiguity.

Response:

We have revised the whole manuscript carefully and tried to avoid any grammar or syntax error. After careful consideration, we have revised the title of the manuscript. The new title is as follows.

Structure and Property of Propellant based on Nitroglycerine/Glycerol Triacetate Mixed Plasticizers: MD Simulation and Experimental Study

2. Authors has done a lot of molecular dynamics simulations, but there are not enough actual experimental results to corroborate the simulation results. Such as combustion and mechanical properties.

Response:

In this paper, MD simulations were performed to investigate the influence of NG/GTA mixed plasticizers on the plasticizing ability of NC binder in solid propellant. Generally, the plasticizing ability of plasticizer affects the performances of combustion, mechanics and energy of propellants. The results suggested that NG/GTA mixed plasticizers had better ability to plasticize nitrocellulose, could enhance the combustion efficiency of propellants effectively at low pressures. Besides, the mechanical properties determined from the MD simulation and experimental study suggested that the ductility of propellant based on NG/GTA mixed plasticizers was improved, implying that NG/GTA mixed plasticizers is higher plasticizing efficiency on NC. These experimental results coincide with the MD simulation.

3. According to the description in the manuscript, mixed plasticizers can help improve the properties of the propellant. But there is no comparative experiment. There is no specific experimental data to prove the significance of adding triacetin. A comparison can lead to more convincing conclusions.

Response:

Four different propellant formulations with different content of NG and GTA were designed as follows Table 1 and prepared. For investigation of the influence of NG/GTA mixed plasticizers on the plasticizing ability of NC binder, the NC/NG and NC/NG/GTA blends were constructed as presented in Fig. 2. The mechanical properties determined from the MD simulation and experimental study suggested that the ductility of propellant based on NG/GTA mixed plasticizers was improved, implying that NG/GTA mixed plasticizers is higher plasticizing efficiency on NC. The burning rates of propellants containing NG/GTA mixed plasticizers were higher than of propellants containing NG plasticizer at low pressure. The combustion platform of propellant containing NG/GTA mixed plasticizers moved to medium

pressure zone. The results suggested that NG/GTA mixed plasticizers had better ability to plasticize nitrocellulose, could enhance the combustion efficiency of propellants effectively at low pressures.

Table 1. Main ingredients of the CMDB propellants

Compositions	Content (wt.%)			
	F-1	F-2	F-3	F-4
NC	22.35	22.35	22.35	22.35
RDX	38	38	38	38
NG	30	22.5	20	15
GTA	-	7.5	10	15
Catalyst and other additives	9.65	9.65	9.65	9.65

Fig. 2. Molecular models of NC/NG and NC/NG/GTA blends

4. In the statement of the conclusion part, specific experimental data is needed to support the author’ s conclusion.

Response:

We have revised the conclusion of the manuscript as follows.

(1) The radial distribution function indicated that NC/plasticizers blends showed stronger intermolecular interaction of van der Waals force and hydrogen bonds. The binding energy of NC/NG blends was 1242.80 kcal/mol, higher than that of NC/NG/GTA blends (759.48 kcal/mol).

(2) The mean-squared displacements of plasticizers revealed that the diffusion coefficients (D) of GTA plasticizer in the NC binder was 6.07×10^{-11} m²/s, higher than that of

NG in the NC binder ($5.83 \times 10^{-11} \text{ m}^2/\text{s}$). The free volume V_{free} of NC/3NG/GTA blend was greater than that of NC/NG blend, which was beneficial to the structural ductility.

(3) Mechanical properties determined from the MD simulation confirmed that the NG/GTA mixed plasticizers increased the elasticity, ductility, and tenacity of the NC. Meanwhile, the mechanical properties tests displayed that the elongation at break of the propellant based on NG/GTA mixed plasticizers was significantly higher than that of the propellant with NG plasticizer at the same temperature.

(4) The burning rates of propellants containing NG/GTA mixed plasticizers were higher than of propellants containing NG plasticizer at low pressure. The combustion platform of propellant containing NG/GTA mixed plasticizers moved to medium pressure zone. The combustion wave temperature demonstrated that the combustion temperature in gaseous phases was less affected by pressure.

In general, these experimental results coincide with the MD simulation. NG/GTA mixed plasticizers had better ability to plasticize nitrocellulose, could enhance the combustion efficiency of propellants effectively at low pressures, facilitated the heat feedback to propellant. The energy, combustion and mechanical properties should be considered comprehensively in the formulation design of high-performance propellant.

Please feel free to contact us if you have any questions about the manuscript.

Thank you for your consideration. I look forward to hearing from you.

Sincerely yours

Ms. Lilong Yang

Xi'an Modern Chemistry Research Institute

Xi'an 710065, China

Tel: 86-029-88294031

E-mail: yanglilong.1988@163.com